# Fabrication of Flexible Wiring with Intrinsically Conducting Polymers Using Blue-Laser Microstereolithography

**DOI:** 10.3390/polym14224949

**Published:** 2022-11-16

**Authors:** Mai Takenouchi, Masaru Mukai, Taichi Furukawa, Shoji Maruo

**Affiliations:** Faculty of Engineering, Yokohama National University, Yokohama 240-8501, Japan

**Keywords:** microstereolithography, flexible wiring, 3D microfabrication, conductive polymers

## Abstract

Recently, flexible devices using intrinsically conductive polymers, particularly poly(3,4-ethylenedioxythiophene) (PEDOT), have been extensively investigated. However, most flexible wiring fabrication methods using PEDOT are limited to two-dimensional (2D) fabrication. In this study, we fabricated three-dimensional (3D) wiring using the highly precise 3D printing method of stereolithography. Although several PEDOT fabrication methods using 3D printing systems have been studied, few have simultaneously achieved both high conductivity and precise accuracy. In this study, we review the post-fabrication process, particularly the doping agent. Consequently, we successfully fabricated wiring with a conductivity of 16 S cm^−1^. Furthermore, flexible wiring was demonstrated by modeling the fabricated wiring on a polyimide film with surface treatment and creating a three-dimensional fabrication object.

## 1. Introduction

Intrinsically conductive polymers (ICPs) have been actively investigated since the discovery of conductive polyacetylene with iodine doping by Shirakawa et al. in 1977 [1], and various ICPs have been developed [2]. In particular, poly(3,4-ethylenedioxythiophene) (PEDOT) is one of the most widely employed ICPs because it is superior to ICPs in terms of conductivity and environmental stability after doping [3,4,5]. Recently, PEDOT has become a promising conductive material in the field of flexible electronics [6,7,8,9,10,11,12,13,14,15], where high processability is required for the advancement of the Internet of Things. Currently, inkjet printing [11], screen printing [12], roll-to-roll method [13], and photolithography [6,7,8,9,10] are the main processing technologies for fabricated wiring. However, these processing technologies have limitations in achieving three-dimensional (3D) wiring and fabrication accuracy. High device integration is effective for fabricating highly functional flexible devices. If 3D flexible wiring can be formed in high definition, device integration can be drastically improved.

As an example of attempted 3D wiring using PEDOT, a 3D printing method has been reported employing material extrusion (MEX) of a solution containing a mixture of PEDOT and polystyrene sulfonic acid (PSS) [16]. However, the MEX method is unsuitable for high-resolution patterning because the resolution of the MEX method depends on the nozzle diameter. In this study, we focused on stereolithography, which is the most precise 3D printing technique. The color that originates from a long-conjugated system of PEDOT is a challenge for adapting PEDOT to the photocurable resin used in stereolithographic methods because the absorption of light suppresses the photopolymerization reaction by PEDOT. Therefore, two methods are used for the 3D-wiring fabrication of PEDOT using stereolithography: one is to fabricate a mold and soak it with a PEDOT:PSS aqueous solution, and the other is to mix the PEDOT precursor with a light-curing resin. For example, in the previous method, Tao et al. developed arbitrary structures from a hydrogel by stereolithography and then immersed the fabricated object in a PEDOT:PSS solution to fabricate wiring [17]. However, it was difficult to achieve sufficient conductivity using only hydrogel and PEDOT:PSS, and although the conductivity was improved by adding a small amount of highly conductive carbon nanotubes to the light-curing resin that formed the hydrogel, the conductivity was only 0.425 S cm^–1^. As an alternative method to fabricate conductive 3D wiring using PEDOT precursors, Yamada et al. impregnated the EDOT dimer in a Nafion film and succeeded in obtaining a high conductivity of 3500 S cm^–1^; however, their method is limited to inner Nafion film [18].

In contrast, Kurselis et al. fabricated a 3D object in which PEDOT is dispersed inside the 3D-printed object by the oxidative polymerization of EDOT with iron chloride (FeCl_3_) inside the fabricated object [19]. In their study, advanced 3D microstructures, such as lattices, were fabricated using two-photon lithography, exhibiting an ultra-high resolution of several hundred nanometers. Furthermore, they suggested that flexible wiring could be achieved by selecting a highly flexible polymer as the matrix. However, the maximum conductivity they achieved was 0.04 S cm^–1^. Although the choice of dopant is important for the conductivity of ICPs [20], they employed FeCl_3_ as both the oxidant for EDOT and the dopant for PEDOT in their experiments. We hypothesized that they could not obtain sufficient conductivity in their study because FeCl_3_ was unsuitable as a dopant. The most common dopant for PEDOT is PSS, and this mixture is termed as PEDOT:PSS. It is obtained by the oxidative polymerization of EDOT dispersed in a PSS solution [3] but applying the same method to the photocurable resin used in stereolithography is difficult. This is because significant amounts of water must be removed after fabrication, which causes shape deformation owing to shrinkage, and the low solubility of EDOT in water causes it to separate from the 3D-printed structures. In addition, it is difficult to dissolve PSS in photocurable resins containing EDOT without a solvent. Even if PSS is doped after oxidative polymerization, the large molecular size of PSS prevents it from penetrating the 3D-printing material, making it difficult to expect doping effects. Hence, we concluded that it would be difficult to use PSS and employed *p*-toluene sulfonic acid (PTSA) as the dopant instead of PSS. The chemical structure of PTSA is similar to the monomer unit of PSS. PTSA possesses a low molecular weight of 172 g mol^–1^ and is expected to exhibit significant penetration into the 3D printing material. Furthermore, it has been reported to have dopant effects on PEDOT [21,22]. In this study, we attempted to improve conductivity by doping with PTSA following the scheme in Figure 1. Consequently, we succeeded in improving the conductivity by approximately two orders of magnitude, even though the EDOT content was less than half the amount reported by Kurselis et al. In stereolithography-based methods for fabricating wiring, most of the reported conductivity is less than 0.01 S cm^−1^, even when metallic particles or carbon nanotubes are used [23,24,25,26,27]. Therefore, it has advantages in terms of conductivity over other materials.

Furthermore, because conductive polymers are dispersed within flexible polymer materials, flexible wiring was demonstrated on a polyimide (PI) film, which was applied as an original surface treatment to improve the adhesion of 3D-printed structures. PI film is widely used as a base material for flexible substrates.

## 2. Materials and Methods

### 2.1. Materials

*p*-Toluene sulfonic acid monohydrate (PTSA) and 3,4-ethylenedioxythiophene (EDOT) were purchased from Tokyo Chemical Industry Co., Ltd. (Tokyo, Japan). Iron (III) chloride hexahydrate (FeCl_3_·6H_2_O), sulfuric acid (H_2_SO_4_), chloroform, ethanol, and branched polyethyleneimine (PEI, average molecular weight: 1800 g mol^–1^) were obtained from FUJIFILM Wako Pure Chemical Co. (Osaka, Japan). Diphenyl(2,4,6-trimethylbenzoyl)phosphine oxide (TPO) was purchased from Sigma–Aldrich (St. Louis, MO, USA). Polyethylene glycol (600) dimethacrylate (SR252) and dipentaerythritol pentaacrylate (SR399) were purchased from TOMOE Engineering Co., Ltd. (Okayama, Japan). A polyimide (PI) film was obtained from 3M Japan Co., Inc. (Tokyo, Japan) Gold wire (diameter: 0.1 mm) was obtained from Nilaco Co. (Tokyo, Japan). Conductive adhesive (Dotite D-500) was purchased from Fujikura Kasei Co., Ltd. (Tokyo Japan). A blue light-emitting diode (LED, EIL33-3L) was obtained from Toyoda Gosei Co., Ltd. (Aichi, Japan).

### 2.2. Characterization

Attenuated total reflectance Fourier-transform infrared (ATR-FTIR) spectroscopy was performed using an FTIR-6200 with ATR PRO450-S (Diamond prism) at 25 °C (JASCO Co., Tokyo, Japan). The conductivity of the 3D models was calculated using the resistance and size of the object. A gold wire was attached to the fabricated object using conductive adhesive. The resistance value of the object was then measured using an Agilent 34410A 6 ½ digital multimeter (Agilent Technologies, Inc., Santa Clara, CA, USA.) with a four-terminal sensing mode. The observation and size evaluation of the fabricated objects were performed using a VHX6000 microscope (Keyence Corp., Osaka, Japan). Scanning probe microscopy (SPM) observations (tapping mode) were performed using an SPA400 (Seiko Instrument Inc., Chiba, Japan) equipped with an SI-DF20(Al) cantilever with a spring constant of 15 N m^−1^. SPM phase-difference and topographic images were analyzed using Gwyddion 2.58.

### 2.3. Stereolithography System

We previously developed several types of stereolithography systems, such as a bottom-up system based on single-photon polymerization with a blue laser [28,29] and a direct writing system using blue and/or femtosecond lasers [30,31]. In this study, we developed another bottom-up system based on single-photon polymerization using a blue laser with an fθ lens for laser writing in centimeter-sized areas. Figure 2 shows a schematic of the developed blue-laser stereolithography system. Laser light is emitted from a semiconductor laser (Cobolt 06-MLD, Cobolt, Solna, Stockholms Lan, Sweden) with a wavelength of 405 nm, and the laser output is adjusted by passing it through a variable neutral density (ND) filter. The laser was turned on and off using an automatic shutter. The laser diameter was then increased using a beam expander. Next, the laser light that passed through the beam splitter cube and entered the observation optics was incident on a galvanometer mirror (SCANCUBE III 14, SCANLAB GmbH, Puchheim, Germany) and focused on the boundary surface between the light-curing resin and upper glass substrate using an objective lens. The system employed an fθ lens (SCANCUBE III 14, SCANLAB GmbH, Puchheim, Germany) as the focusing lens. The laser light reflected by the galvanometer mirror can only form an image along a circular surface using a spherical lens. However, fθ lenses overcome this limitation by flattening and correcting the image plane of the laser light. This system enables scanning on a flat surface over a larger area than spherical lenses and is effective for laser wiring. The build area of the system was 60 × 60 mm^2^. Based on the slice data, the galvanometer mirror was scanned in the horizontal plane to cure the light-curing resin and form the layers. One portion of the laser light reflected by the glass-bottom Petri dish was reflected by a beam splitter cube and focused on a charge-coupled device camera to precisely adjust the focus position onto the glass substrate. The 3D model was formed upside-down on the upper glass substrate, and the Z stage was raised to stack each layer. These stages and galvanometers automatically controlled the laser writing, stacking, and scanning speed changes according to the 3D-CAD model data using lab-made software to fabricate the 3D model. The irradiated laser outputs for 1, 2, 3, 4, 5, 6, 7, 8, 9, 10, 17, and 25 vol.% of EDOT in uncured resin were 0.25, 0.25, 0.25, 0.25, 0.25, 0.255, 0.255, 0.26, 0.26, 0.26, 0.30, and 0.30 mW, respectively, when passing through the variable ND filter in our setup. The layer pitch and hatching distances were 30 and 3 μm, respectively.

### 2.4. Preparation of Resin for Conductive Fabrication Objects

For all resins mixed with EDOT and SR252 in different volume ratios, and a TPO initiator was added at 1 wt.% to SR252. The mixture was stirred for 5 min.

## 3. Results and Discussion

### 3.1. Conductive Treatment of Resin Droplet

First, the formation of PEDOT, a conductive polymer, in the cured resin was confirmed. In this study, ATR-FTIR was employed to confirm the synthesis of PEDOT from the changes in the chemical structure of the cured material. In this case, resin droplets containing 44.8 vol.% EDOT were adjusted as a model system, and curing was attempted using ultraviolet (UV) irradiation (handheld UV lamp SLM-8, wavelength: 365 nm, 1407 µW cm^–2^, AS One Co., Osaka, Japan). The oxidative polymerization of EDOT with FeCl_3_·6H_2_O was then performed (Figure 3). In addition, a higher concentration of EDOT was used to facilitate the confirmation of the spectral changes associated with the oxidative polymerization of EDOT.

The ATR-FTIR spectra of the uncured resin, photopolymerized resin UV-irradiated for 15 min, and oxidative polymerization resin treated with FeCl_3_·6H_2_O are shown in Figure 4. In all the spectra, peaks were observed at 2859 cm^–1^, attributed to the symmetrical stretching vibration of CH_2_ with SR252, and 1712 cm^–1^, attributed to the carbonyl group (C=O) stretch vibration. The peak originating from SR252 appears strong because SR252 accounts for a large proportion of the resin composition. In the uncured resin, the 1624 cm^–1^ peak originating from SR252 acrylate (C=C stretch vibration) was observed (Figure 4, bottom spectra), but the peak decreased significantly after photopolymerization (Figure 4, middle spectra) and oxidative polymerization (Figure 4, upper spectra). This means that the acrylate groups disappeared owing to photopolymerization. The adjusted uncured resin containing EDOT was a clear liquid (Figure 5a). However, after UV irradiation, it was a transparent solid, which can also be explained by the fact that SR252 formed a cross-linked structure following photopolymerization (Figure 5b). Further, it remains transparent, suggesting that EDOT has not polymerized. The 754 cm^–1^, 892 cm^–1^, and 1185 cm^–1^ peaks were derived from the C-H stretch vibration, out-of-plane vibration of =C-H, and in-plane vibration of =C-H in EDOT, respectively [32]. This peak was observed in the uncured resin (Figure 4, bottom spectra) and sample after photopolymerization (Figure 4, middle spectra) and nearly disappeared after oxidative polymerization (Figure 4, upper spectra). Because EDOT is not involved in photopolymerization and the solid after photopolymerization exhibits almost no liquid release, it can be designated as an organogel with EDOT as the solvent (Figure 5b). The decrease in those peaks’ intensity after oxidative polymerization is presumably due to the loss of hydrogen atoms (α-α’ coupling) on the thiophene ring associated with PEDOT formation [32,33]. The formation of PEDOT can also be explained by the fact that the solid after photopolymerization is transparent (Figure 5b), whereas after oxidative polymerization, it is a black solid (Figure 5c), which is due to the red shift of the light absorption region owing to the increased conjugation length caused by the polymerization of EDOT [34]. In addition, as a simple continuity test, a sheet resistance of 0.38 ± 0.29 kΩ/sq. was confirmed using a resistivity meter (Loresta-GX, MCP-T700 with Loresta probe type PSP MCP-TPQP2). These results indicate that the solids were materials containing PEDOT as a conductive polymer. Furthermore, the polymerized solid of SR252 is soft and forms a flexible film. PEDOT is a non-flexible solid; however, the high proportion of SR252 in the resin confirms that it is sufficiently flexible to bend with tweezers (Figure 5d).

SPM investigated the reasons for obtaining flexible and conductive materials. SPM in the tapping mode was used to record topographical (Figure 6, top) and phase-difference images (Figure 6, bottom). The topographical image provides information on surface roughness, whereas the phase-difference image reflects the viscoelastic properties of the surface. In the topographic image, the height difference was approximately 10 nm after photopolymerization, whereas the height difference was approximately 300 nm after oxidative polymerization. This is suggested to be due to the shrinkage of the structure caused by oxidative polymerization, resulting in wrinkles on the surface. The phase-contrast image after photopolymerization is used as a uniform material, as there are negligible differences in the shading. However, after oxidative polymerization, dark and light areas were clearly visible. This suggests that the two layers had separate structures [29]. In this case, polymerized SR252 provides a soft and flexible material. In contrast, PEDOT is an amorphous solid and possesses superior viscoelastic properties compared with polymerized SR252. Therefore, areas with a high percentage of SR252 in the phase-contrast images were considered bright, whereas those with a high percentage of PEDOT were dark. The SPM results indicate that SR252 acts as a matrix and is flexible, whereas PEDOT is dispersed within the matrix at the nanoscale and possesses a continuous structure, which is considered conductive.

### 3.2. Tuning of the Resin Composition to Improve Conductivity with PTSA Doping

The effect of doping the fabricated object with PTSA was evaluated based on conductivity, and the optimal conditions were determined by varying the ratios of SR252 and EDOT in the resin. The fabrication and post-curing processes were performed using a method similar to that described by Kurselis et al. [19]. The difference between this study and that of Kurselis et al. is that the oxidative polymerization and doping process were performed in two steps in this study, instead of one. Although further efforts are required, the conductivity is expected to improve using an appropriate compound in each step. Figure 1 shows the stereolithographic fabrication and post-curing process. First, any structure was fabricated on an acrylate-modified glass substrate from the prepared resin using a blue-laser stereolithography system. The EDOT in the 3D-printed object was then polymerized to PEDOT by oxidative polymerization in melted FeCl_3_·6H_2_O at 120 °C for 7 min. Doping was then performed by immersion in a PTSA aqueous solution (5.2 M) for 10 min. After washing, the samples were dried in a vacuum desiccator for at least 24 h (<0.1 kPa). In addition, we considered mixing PTSA with EDOT in photocurable resin in advance. However, although a clear, colorless liquid could be prepared immediately after adjustment, it turned blue-black and discolored after several days of standing. This is believed to be because PTSA causes the polymerization of EDOT, even in the absence of an oxidizing agent [35]. Oxidative polymerization followed by doping was employed to prevent coloration before the material was cured.

To evaluate the conductivity of the fabricated wiring, terminals consisting of 1 mm^2^ squares were fabricated at both ends of a 5 mm long, 0.5 mm wide wire (Figure 7a, insert). Figure 7a shows the wiring pattern fabricated using blue-laser stereolithography and oxidative polymerization. The fabricated pattern was confirmed to be shaped as designed. In addition, the color of the fabricated pattern changed to black, owing to the formation of PEDOT by oxidative polymerization. To evaluate the conductivity, a gold wire was connected to the fabricated wire using a conductive adhesive (Figure 7b). In the absence of PTSA doping, an increase in the percentage of EDOT exhibited a logarithmic increase (Figure 8, open circle). Because the EDOT content increases, the conductive component, PEDOT, increases after oxidative polymerization. In contrast, the PTSA-treated samples exhibited improved electrical conductivity in all samples below 20 vol.% EDOT, suggesting the superiority of PTSA over FeCl_3_ as a dopant. Furthermore, the samples doped with PTSA exhibited a maximum conductivity of approximately 5–8 vol.% of EDOT content in the uncured resin (Figure 8, closed circle). The fabricated object was immersed in an aqueous solution of PTSA. PEDOT is insoluble in water; however, SR252 is hydrophilic; therefore, doping is believed to proceed by the penetration of water with dissolved PTSA between the polymer network of SR252; that is, SR252 acts as a dopant channel. As the percentage of EDOT increased, the percentage of the SR252 that acted as a flow channel decreased; therefore, the effect of the dopant was likely to decrease as the percentage of EDOT increased above a certain level. In addition, an increase in the EDOT content implies a decrease in the content of SR252, which is responsible for shape retention, as well as an increase in the amount of PEDOT, which is inflexible, making the fabricated structure brittle [36]. In the Kurselis system, which is not doped with PTSA, 17 vol.% EDOT is the limit for shape retention and high conductivity (0.04 S cm^–1^) [19]. However, our system doped with PTSA exhibits a conductivity more than 100 times higher than that of our system with 5–8 vol.% EDOT (maximum 16 S cm^–1^ at 7 vol.% EDOT), which is less than half the amount. This would be an effective means of achieving both conductivity and structural stability such as flexibility.

### 3.3. Flexible Wiring Using PI Film as a Substrate

We attempted to construct flexible wiring on a PI film. However, the adhesive strength of the fabricated object on the pristine PI film was low, and the film peeled off. In the case of flexible wiring, if the adhesive strength to the substrate is insufficient, delamination may occur owing to curvature; therefore, the adhesive strength between the substrate and the fabricated object is important [37,38]. To improve the adhesion between the PI film and the fabricated object, the surface of the PI film was acrylated using PEI and SR399 (PI-PEI-SR399). Because this film has acrylate groups on its surface, it is expected to exhibit a strong adhesive strength because the acrylate groups on the substrate are also polymerized during the photopolymerization process of the 3D object [39].

Figure 9 illustrates the surface treatment process. First, the PI film was immersed in 98% H_2_SO_4_ for 1 min at 25 °C to modify its surface. The ATR-FTIR spectrum at the bottom of Figure 10 (orange) shows the pristine PI film. Peaks at 1775 and 1713 cm^–1^ from the imide group (asymmetrical and symmetrical C=O stretch vibration), 1509 cm^–1^ from the aromatic C=C stretch vibration, 1382 cm^–1^ from the C–N stretching vibration of the imide group, and 1241 cm^–1^ from the aromatic ester (C–O–C) were observed in the pristine PI film [40]. After the H_2_SO_4_ treatment of the PI film, a new peak was observed at 1199 cm^–1^, indicating the modification with the sulfonate group (symmetric stretch vibration of SO_2_) on the surface of the PI film (Figure 10, black). Similar to PI films, polymers with aromatic esters have been modified with sulfate groups by H_2_SO_4_ treatment at 25 °C [41]. The product was then immersed in an ethanol solution of PEI (PEI:ethanol = 1:3 (*w*/*w*)) for 1 min. The sulfonate group present on the surface of the PI (H_2_SO_4_ treatment) film was anionic, whereas the PEI was electrostatically adsorbed because of its cationic amine groups. The excess PEI was then washed with water and ethanol. The PI–PEI film exhibits a peak of amine groups (C–N vibration) derived from adsorbed PEI observed at 1044 cm^–1^ (Figure 10, blue). The sample was then immersed in a chloroform solution of SR399 (SR399:chloroform = 1:7 (*w*/*w*)) for 30 min. SR399 is a multifunctional monomer containing five acrylate groups. The acrylate group of SR399 and amine group of PEI formed a bond by the Michael addition reaction; however, because excess SR399 was added to PEI, most of the SR399 acrylate remained unreacted. After washing with chloroform and ethanol, the peaks at 1624 and 1395 cm^–1^ derived from the SR399 acrylate group (CH=CH_2_ stretch vibration and CH=CH_2_ scissor bending vibration, respectively), and the 1200 cm^–1^ peak derived from the acrylate ester group (C–O–C stretch vibration) of SR399 were observed by ATR-FTIR, confirming surface acrylation (Figure 10, green).

The fabricated wiring was formed on the PI-PEI-SR399 film, which was able to form the wiring without peeling the film (Figure 11a). The results indicated that the PI-PEI-SR399 film successfully improved the adhesion between the fabricated object and the film. Furthermore, when an LED and power source (PW18-3AD, Kenwood Co., Tokyo, Japan) were connected in series with the curved fabricated object and voltage was applied, the LEDs emitted light (Figure 11b). This clearly indicates that the fabricated wiring is conductive, even on the bent PI-PEI-SR399 film, and can function as flexible wiring.

Up to now, the 2D modeling for evaluating the properties of adjusted resins has been relatively simple. As a next step, geometric modeling was attempted as a more complex 2D modeling pattern (Figure 12a). In addition, two towers, a bridge, and a stepped pyramid were modeled as 3D modeling objects (Figure 12b–d). Both moldings indicated that 2D and 3D structures could be built.

## 4. Conclusions

In this study, a photocurable resin was prepared by mixing SR252 and EDOT, a PEDOT precursor, as a flexible polymer matrix. This confirmed that an organogel with EDOT as a solvent was formed by cross-linking SR252 by photopolymerization. ATR-FTIR analysis confirmed that the oxidative polymerization of EDOT by FeCl_3_·6H_2_O occurred in the cured material to form conductive structures encapsulating PEDOT. It was also confirmed that the solid formed by these reactions was a flexible material and did not fracture when bent. The wiring was then formed using a laboratory-constructed microstereolithography system with a blue laser, and the effect of dopant treatment with PTSA was evaluated based on conductivity. Consequently, it was confirmed that PTSA is a superior dopant compared to FeCl_3_·6H_2_O, and the conductivity was improved more than 100 times despite the EDOT ratio being less than half that of Kurselis et al. [19]. Furthermore, it was determined that by using this resin, the 3D fabrication of conductive objects is possible. When LEDs were connected to the surface-treated PI film with wiring fabricated using this method, the light continued to shine even when the film was bent, indicating that the fabricated wiring could function as flexible wiring. The proposed method provides a flexible conductive 3D object using a stereolithography system, thereby allowing the fabrication of sophisticated flexible devices. In addition, we are planning to add inorganic materials with high conductivity as a way to further improve the conductivity [17,42].

## Figures and Tables

**Figure 1 polymers-14-04949-f001:**
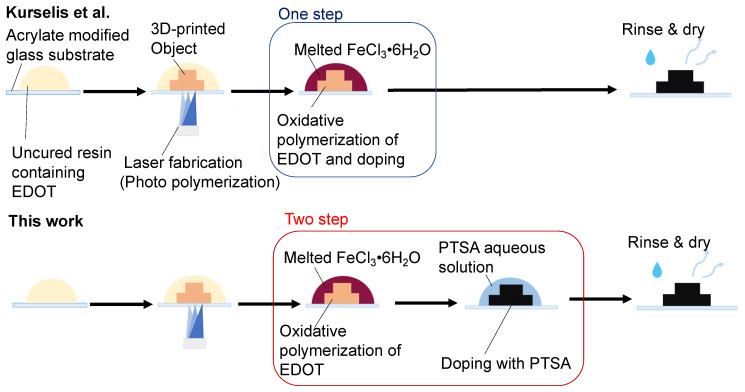
Schematic of different post-curing processes between Kurselis et al. [19] and this study.

**Figure 2 polymers-14-04949-f002:**
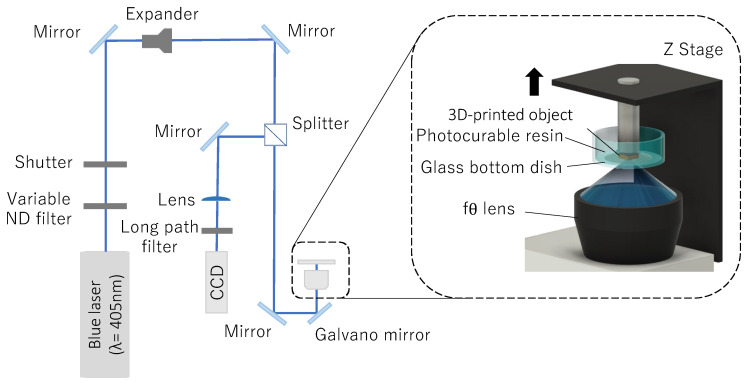
Schematic of the bottom-up stereolithography system using a blue laser.

**Figure 3 polymers-14-04949-f003:**
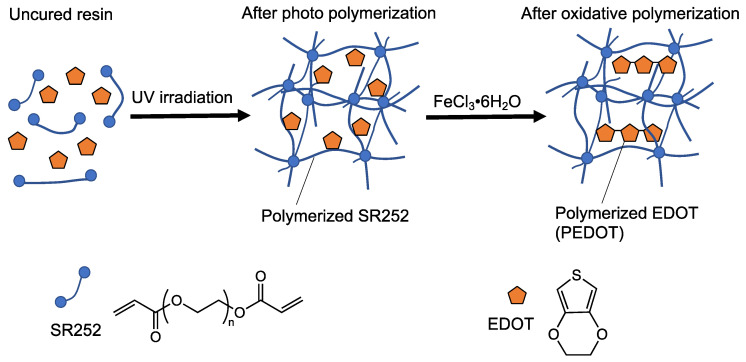
Schematic of expected chemical structure change for each process.

**Figure 4 polymers-14-04949-f004:**
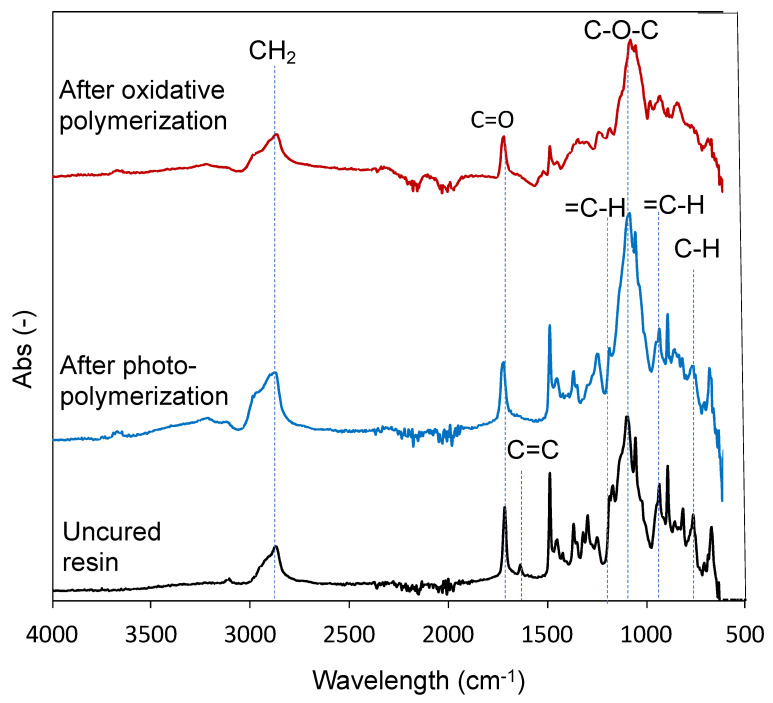
ATR-FTIR spectra of the uncured resin containing EDOT, after the photopolymerization of the resin, and after oxidative polymerization of the resin.

**Figure 5 polymers-14-04949-f005:**
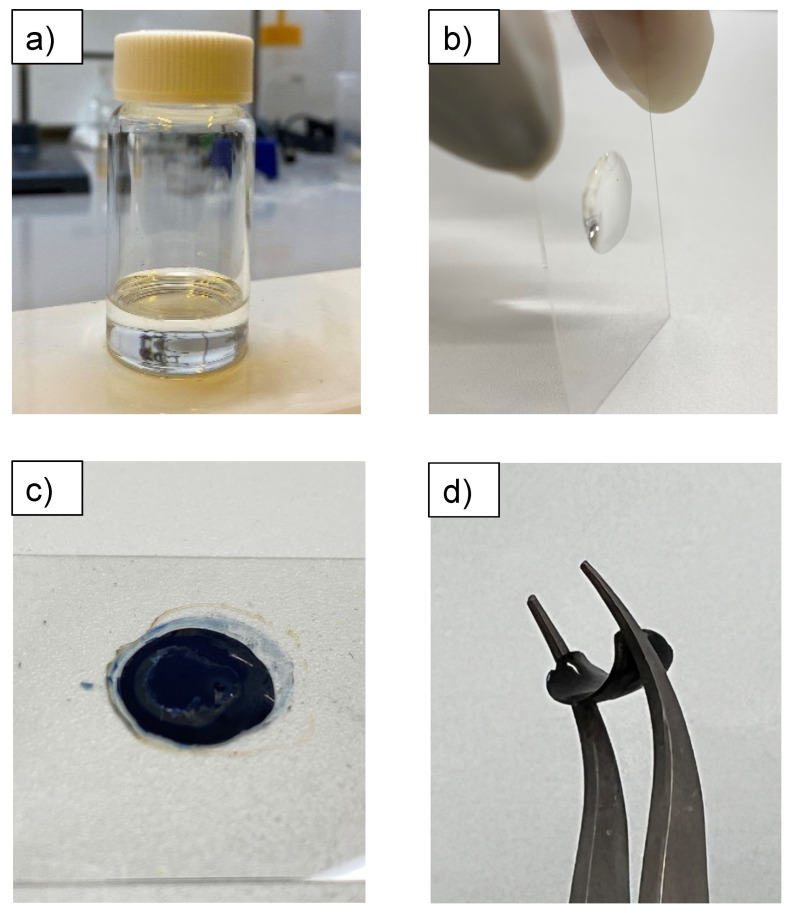
Photographs of the uncured resin containing EDOT (**a**); resin cured by photopolymerization (**b**); resin after oxidation polymerization (**c**); and fabricated flexible sample bent with tweezers (**d**). The diameter of the cured droplet was 20 mm.

**Figure 6 polymers-14-04949-f006:**
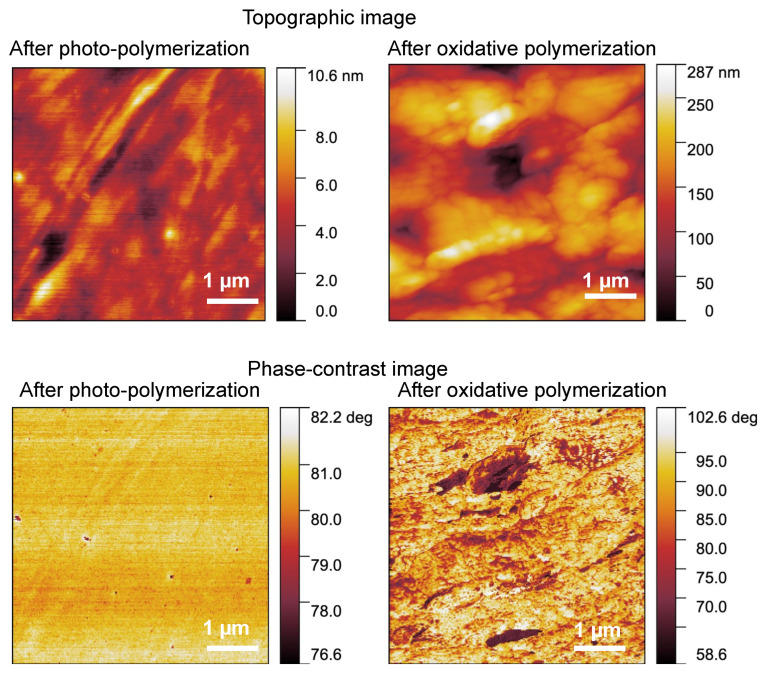
SPM topography (**top**), and phase-contrast (**bottom**) images of fabricated objects.

**Figure 7 polymers-14-04949-f007:**
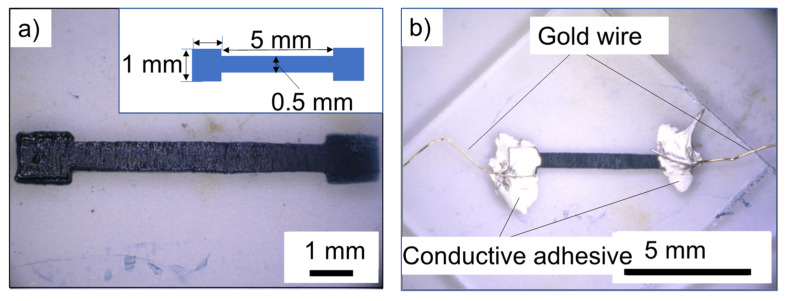
Photograph of an example of fabricated wiring (inserted picture indicates model) (**a**); and gold wire with conductive adhesive to evaluate conductivity of fabricated wiring model (**b**).

**Figure 8 polymers-14-04949-f008:**
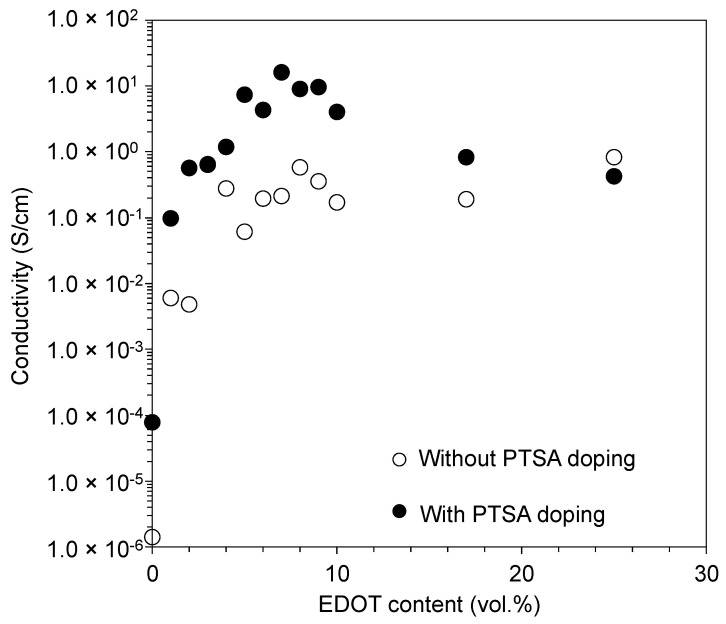
Electrical conductivity of fabricated wiring before (open circles) and after (closed circles) PTSA doping.

**Figure 9 polymers-14-04949-f009:**
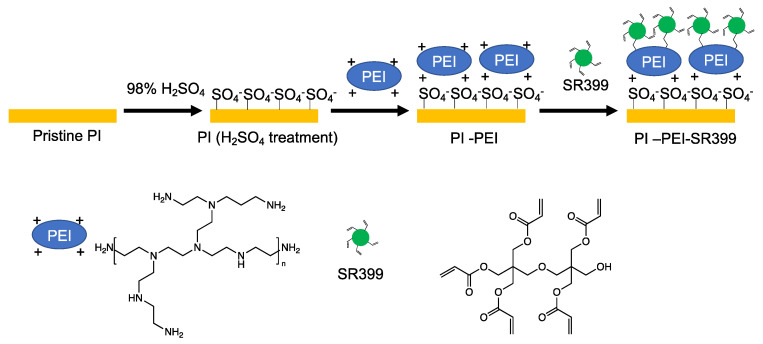
Schematic of surface arylation by PEI on PI.

**Figure 10 polymers-14-04949-f010:**
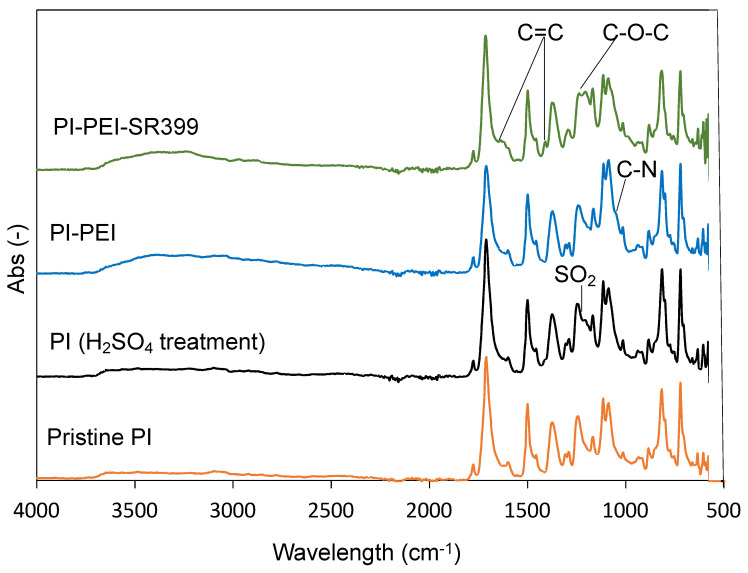
ATR-FTIR spectra of pristine PI, H_2_SO_4_-treated PI, PI-PEI, and PI-PEI-SR399.

**Figure 11 polymers-14-04949-f011:**
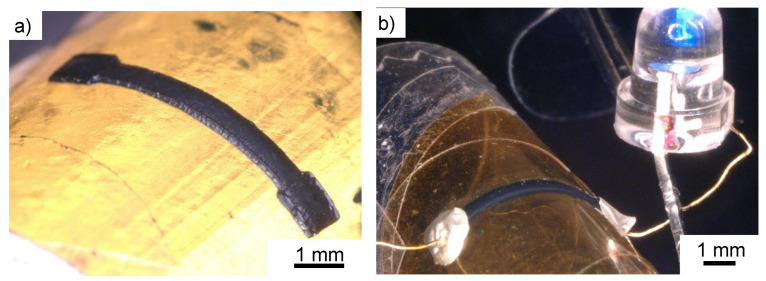
Example of fabricated flexible wiring on PI-PEI-SR399. (**a**) Flexible wiring on PI-PEI-SR399. (**b**) Blue LED and power source connect fabricated flexible wiring on PI-PEI-SR399.

**Figure 12 polymers-14-04949-f012:**
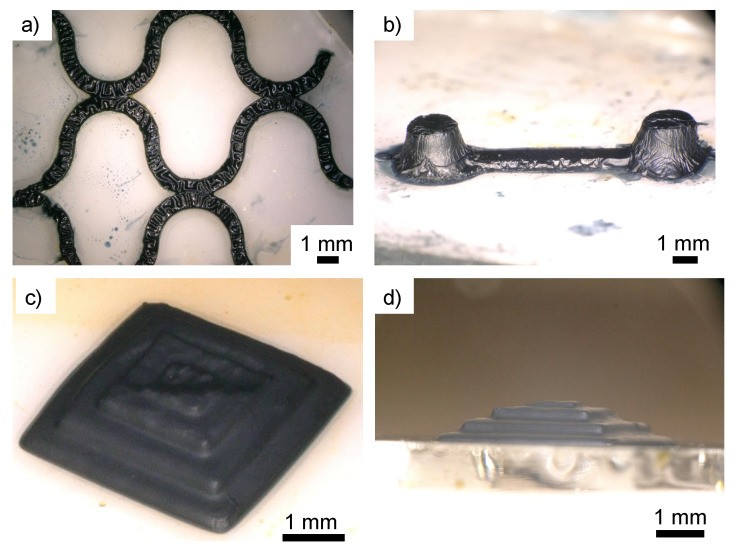
Examples of fabricated 2D and 3D structures. (**a**) 2D geometric model; (**b**) 3D two towers and a bridge model; (**c**) bird’s eye view of 3D stepped pyramid; and (**d**) side view of 3D stepped pyramid. Resin with 5% EDOT was used for fabrication. Resin with 5% EDOT was used for modeling; 2D and 3D structures were modeled with laser intensities of 0.2 and 0.04 mW, respectively.

## Data Availability

The datasets generated and/or analyzed during the current study are available from the corresponding author on reasonable request.

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
