# Peer review of "Fabrication of Flexible Wiring with Intrinsically Conducting Polymers Using Blue-Laser Microstereolithography"

_polymers, 2022, doi:10.3390/polym14224949_

Round 1

Reviewer 1 Report

Recommendation Acceptance: Minor revision.

Comment: 

This manuscript was prepared about the new method to fabricate the conductive structures comprised of PEDOT in a stereolithography system. In addition, it was found that the manufactured wiring may function as a flexible wiring, and using this, it is possible to manufacture a sophisticated flexible device. Overall the manuscript is well-organized and results are convincing. I would like to recommend the publication of this manuscript after addressing the following minor issues.

1.      The authors need to adjust the captions inserted in figures more visible format.

2.      The word 'dopant' on line 59 on page 2 needs an article. Typo and grammar should be checked through the manuscript.

Reviewer 2 Report

1.       No citations in the abstract. Rephrase line 13.

2.       The study has been repeatedly cross-examined with the work of Kurselis et al. This impacts the strength of the work and reduces its scope to only comparative research – not helpful to the community. Remove or generalize the comparisons widely. 

3.       Line 177, the 892 cm1 peak assignment is not correct. It is the C–S stretching vibration of PEDOT thiophene ring (following references). The same peak disappearance has been observed in PEDOT:PSS. Rephrase your reasoning for the decrease in the peak intensity based on the following studies. Also, authors need to cite all their peak assignments in the manuscript.

https://doi.org/10.1039/C9TC06311K

https://doi.org/10.1021/acsami.0c03544

4.        Line 188, firstly, it is the resistance that has been reported not the conductivity. Secondly, report the value rather than writing “several kΩ”.

5.       Have the authors tested the mechanical stability, and strain-stress tests as in other studies?

DOI: 10.1039/d1ma00656h

DOI: 10.1002/adfm.202002853

DOI: 10.1016/j.actbio.2021.07.069

6.       Line 202: “The reason for obtaining flexible and conductive materials was investigated??? using SPM.” The sentence is odd! Consider rephrasing.

7.       Figure 8: Change the Y-axis values to the scientific numbers (10–6 etc.)

8.       Authors need to compare their work with the following recent study using a silver nanowire in combination with PEDOT:PSS as a 3D structure with higher conductivity and explain their advantages:

https://doi.org/10.1016/j.cej.2022.134598

Reviewer 3 Report

Paper describes stereolithographic fabrication of flexible wiring made from PEDOT conductive polymer. Authors report that similar was made in previous research by other authors but they clearly describe what is improved so the novelty is sufficient to justify publication. Research design is appropriate as well as the results presentation, discussion and derived conclusions, cited references are relevant. Paper has couple of smaller drawbacks so minor revision is suggested before the publication.

1)      Line 156-158. It’s not really clear what has been done here. EDOT ratio in your samples was 1-25%, but here you mention 44.8% as model system. Why exactly this percentage, is it to test possibility of curing the resin with higher percentage of EDOT than in your sample? You should clarify it.

2)      It’s not clear from the paper why is it important to have substrate, in this case PI? Please explain and try to connect it with possible application of such system.

3)      Figures 4 and 10. FTIR spectra have high level of overlapping absorptions at the very end of the measurement area (600-500 cm-1) making that part unreadable. I suggest authors to cut that part out to make it more readable as it does not contain useful information. This is probably the results of absorption of IR light by the ATR crystal itself? Also, please add in characterization part which type of crystal is in the used IR instrument.

4)      Line 258. Sentence is “The PTSA was dissolved in water and immersed in the fabricated object for doping”. I guess it should be other way around; fabricated object was immersed in dissolved PTSA?

5)      Line 313-14. This should be SR399, not SR252. Also, surface acrylation was confirmed, not acylation.
